# Fitness of the Papaya Mealybug, *Paracoccus marginatus* (Hemiptera: Pseudococcidae), after Transferring from *Solanum tuberosum* to *Carica papaya*, *Ipomoea batatas*, and *Alternanthera philoxeroides*

**DOI:** 10.3390/insects13090804

**Published:** 2022-09-02

**Authors:** Hui-Yu Chuai, Meng-Zhu Shi, Jian-Yu Li, Li-Zhen Zheng, Jian-Wei Fu

**Affiliations:** 1Institute of Quality Standards & Testing Technology for Agro-Products/Fujian Provincial Key Laboratory of Quality and Safety of Agricultural Products, Fujian Academy of Agricultural Sciences, Fuzhou 350003, China; 2Institute of Plant Protection, Fujian Academy of Agricultural Sciences, Fuzhou 350003, China; 3College of Plant Protection, Fujian Agriculture and Forestry University, Fuzhou 350002, China

**Keywords:** *Paracoccus marginatus*, host plant shifting, two-sex life table, fitness

## Abstract

**Simple Summary:**

The papaya mealybug, *Paracoccus marginatus,* is a polyphagous invasive pest that causes severe damage in China. To improve our understanding of the expansion and prevalence of *P*. *marginatus* individuals on host plants, it is important to explore the fitness changes of insects after host plant shifting. In this study, we measured the development, fecundity, and population parameters in *P*. *marginatus* individuals over a span of three consecutive generations after being transferred from potato (*Solanum tuberosum*) to papaya (*Carica papaya*), sweet potato (*Ipomoea batatas*), and alligator weed (*Alternanthera philoxeroides*). Further, the population growth rates of insects on *C*. *papaya*, *I*. *batatas,* and *S*. *tuberosum* in the F_2_ generation were projected. We found that *P*. *marginatus* individuals transferred to *C*. *papaya* had higher fitness. When transferred to *I*. *batatas*, the fitness of *P*. *marginatus* initially decreased in F_0_ and then rebounded in F_1_ and F_2_. *Paracoccus marginatus* individuals could rapidly expand their populations on the above host plants. However, *P*. *marginatus* individuals were unable to complete their development on *A*. *philoxeroides*. Our findings provide new insights into the host plant fitness, prevalence, and potential pest control of *P. marginatus*.

**Abstract:**

The papaya mealybug, *Paracoccus marginatus* Williams and Granara de Willink (Hemiptera: Pseudococcidae), is a polyphagous invasive pest in China. The effect that the shifting of the host plant has on the fitness of a polyphagous pest is critical to its prevalence and potential pest control. In order to assess the fitness changes of *P*. *marginatus* after transferal from potato (*Solanum tuberosum* (Tubiflorae: Solanaceae)) to papaya (*Carica papaya* (Parietales: Caricacea)), sweet potato (*Ipomoea batatas* (Tubiflorae: Convolvulaceae)), and alligator weed (*Alternanthera philoxeroides* (Centrospermae: Amaranthaceae)), the life table data of three consecutive generations were collected and analyzed using the age-stage, two-sex life table method. The results showed that when *P*. *marginatus* was transferred from *S*. *tuberosum* to papaya, a higher intrinsic rate of increase (*r*) and finite rate of increase (*λ*) were observed. *Paracoccus marginatus* individuals transferred to *I*. *batatas* had the significantly lower population parameters than those on *C*. *papaya*; however, the fitness recovered for those on *I*. *batatas* after two generations. *Paracoccus marginatus* individuals were unable to complete development on *A*. *philoxeroides*. Our results conclusively demonstrate that *P*. *marginatus* individuals can readily adapt to *C*. *papaya* and *I*. *batatas* even after host plant shifting, and are capable of causing severe damage to these hosts.

## 1. Introduction

The papaya mealybug, *Paracoccus marginatus* Williams and Granara de Willink (Hemiptera: Pseudococcidae), is a globally invasive pest, which attacks host plants by sucking sap from the leaves, stems, and other plant parts [1]. Since the 1990s, *P*. *marginatus* had spread rapidly through the Americas, Africa, and most provinces of southern China [2,3]. *Paracoccus marginatus* is a polyphagous pest of many economic crops and weeds included in more than 64 families, such as Euphorbiaceae, Rubiaceae, and Caricaceae [4]. In our preliminary field investigation, *P. marginatus* was found on important field crops including potato (*Solanum tuberosum* (Tubiflorae: Solanaceae)), papaya (*Carica papaya* (Parietales: Caricacea)), and sweet potato (*Ipomoea batatas* (Tubiflorae: Convolvulaceae)). It has also been observed on alligator weed (*Alternanthera philoxeroides* (Centrospermae: Amaranthaceae)). Most of the insects were observed on the leaves of the above plants. *Solanum tuberosum* has been used for a number of years as a host for the mass-rearing of *P. marginatus* in the laboratory.

Polyphagous insects, which characteristically have a wide range of host plants in nature, are often associated with host shifting. Huang et al. (2014) demonstrated that the rapid host shifting of *Phenacoccus solenopsis* (Tinsley) (Hemiptera: Pseudococcidae) was due to its efficient host plant fitness [5]. The fitness of insects on a new host plant can be evaluated from the population growth rate on the new host plant versus their old host plant [6,7]. Therefore, it is important to understand how invasive pests adapt on the new host plant using demographic characteristics.

Many studies have shown that significant effects may occur in an insect species after host plant shifting, such as changes in their development and fecundity. For example, Milanović et al. (2016) reported that when *Lymantria dispar* L. (Lepidoptera: Lymantriidae) transferred from Hungarian oak to Turkey oak, the developmental time shortened while the efficiency of food utilization increased [8]. Mody et al. (2007) demonstrated that host plant shifting had a strong effect on *Chrysopsyche imparilis* (Lepidoptera: Lasiocampidae), especially in adult fecundity and the mean body mass of second-instar larvae [9]. Furthermore, when assessing the effects of host plant shifting based on the life table and population dynamics of *Aphis gossypii* (Glover) (Hemiptera: Aphididae), Fan et al. (2018) showed that the fecundity (*F*), intrinsic rate of increase (*r*), finite rate of increase (*λ*), and net reproductive rate (*R*_0_) significantly increased when transferred from wheat to cotton [10]. Amarasekare et al. (2008) reported the survival rates of *P*. *marginatus* on different host plants [11]. Nisha and Kennedy (2017), using the female age-specific life table, reported the life table results for *P*. *marginatus* on different host plants [12]. However, the results were limited by ignoring the males in the population [13].

We hypothesized that the fitness of *P*. *marginatus* would change after host plant shifting. We also predicted that *P*. *marginatus* can readily adapt to some new host plants within a few generations. To test these hypotheses, we measured the development, fecundity, and population parameters in *P*. *marginatus* over a span of three consecutive generations after being transferred from *S*. *tuberosum* to *C*. *papaya*, *I*. *batatas,* and *A*. *philoxeroides*. Further, we projected the population growth of the insects on *C*. *papaya*, *I*. *batatas,* and *S*. *tuberosum* in the F_2_ generation.

## 2. Materials and Methods

### 2.1. Cultivation of Host Plants

*Carica papaya* (Parietales: Caricaceae), *I*. *batatas* (Tubiflorae: Convolvulaceae), *S*. *tuberosum* (Tubiflorae: Solanaceae), and *A*. *philoxeroides* (Centrospermae: Amaranthaceae) were obtained from the Institute of Plant Protection, Fujian Academy of Agricultural Sciences. The host plants were cultivated (40.0 cm in length, 30.0 cm in width, and 15.0 cm in height) with nutrient soil (Cuijun, Fuzhou, China) and kept in growth chambers (PRX-450D, Haishu Safe Apparatus, Ningbo, China) at 28 ± 1 °C, 70 ± 5% RH, with a photoperiod of 14: 10 (L: D) h. Young leaves (<30-d old) were used for the study.

### 2.2. Paracoccus marginatus

The eggs of *P*. *marginatus* were originally obtained from a papaya orchard in Fuzhou city (Fujian Province, China, 25°15′~26°39′ N, 118°08′~120°31′ E), and reared on the leaves of *S*. *tuberosum* in a growth chamber for 20 generations to allow *P*. *marginatus* to adapt to *S*. *tuberosum* as a host. The female life cycle (which differs from that of the male) consists of the egg, three larval instars, and the adult stage, while the male life cycle includes the egg, three larval instars, a pupal stage, and the adult stage.

### 2.3. Life Table Study of P. marginatus

Egg masses laid within a 24 h period on potato leaves were randomly selected for the life table study. In order to accurately observe the lifespan of each insect, the eggs were placed on leaves of *C*. *papaya*, *I. batatas*, *S*. *tuberosum*, or *A*. *philoxeroides* in plastic dishes (3.5 cm in diameter and 2.0 cm in height) containing agar (3%). After hatching, each 1st instar was transferred into a fresh dish containing leaves of the same plant and reared individually. Following the advice of Mou et al. (2015), only hatched eggs were used in the life table studies to accurately estimate the life table parameters [14]. Newly emerged adult males and females were paired. The daily fecundity and survival were recorded until the death of all individuals. The life table data for three consecutive generations (F_0_, F_1_, and F_2_) were recorded. *Paracoccus marginatus* reared on *A*. *philoxeroides* only survived for a single generation (F_0_); therefore, only one life table could be constructed for insects on this host.

### 2.4. Life Table Data Analysis

The raw life history data of all individuals of *P*. *marginatus*, including the developmental duration, longevity, and female fecundity, were analyzed according to the age-stage, two-sex life table procedure [15,16] using the program TWOSEX-MSChart [17]. The variances and standard errors of parameters were estimated using the bootstrap technique [18,19]. The differences between treatments were assessed using paired bootstrap tests [20]. The age-stage-specific survival rate (*s_xj_*) is the probability that each hatched egg will survive to age *x* and stage *j*. The age-specific survival rate (*l_x_*) was calculated as:$$l_{x}=\overset{k}{\underset{j=1}{\sum }}s_{xj}$$
where *k* is the number of stages. The age-specific fecundity (*m_x_*) was calculated as:$$m_{x}=\frac{\sum_{j=1}^{k}s_{xj}f_{xj}}{\sum_{j=1}^{k}s_{xj}}$$

The intrinsic rate of increase (*r*) was estimated using the Euler–Lotka equation [21,22] with the age indexed from 0 [23]:$$\overset{\infty }{\underset{x=0}{\sum }}e^{-r\left(x+1\right)}l_{x}m_{x}=1$$

The finite rate of increase (*λ*), net reproductive rate (*R*_0_), and mean generation time (*T*) were calculated as follows:$$\lambda =e^{r}$$
$$R_{0}=\overset{\infty }{\underset{x=0}{\sum }}l_{x}m_{x}$$
$$T=\frac{\ln R_{0}}{r}$$

The age-stage-specific life expectancy (*e_xj_*), i.e., the length of time that an individual of age *x* and stage *j* is expected to survive, was calculated according to Chi and Su (2006) [24]: $$e_{xj}=\overset{\infty }{\underset{i=x}{\sum }}\overset{a}{\underset{y=j}{\sum }}s_{iy}^{\prime }$$
where $s_{iy}^{\prime }$ is the probability that an individual of age *x* and stage *j* can survive to age *i* and stage *y* by assuming that $s_{iy}^{\prime }=1$. The age-stage reproductive value (*v_xj_*), which represents the contribution of each individual in age *x* and stage *j* makes to the future population [25,26], was calculated as:$$v_{xj}=\frac{e^{r\left(x+1\right)}}{s_{xj}}\overset{\infty }{\underset{i=x}{\sum }}e^{-r\left(x+1\right)}\overset{\beta }{\underset{y=j}{\sum }}s_{iy}^{\prime }f_{iy}$$

### 2.5. Population Projection

The population growth of *P*. *marginatus* was simulated according to Chi (1990) [27] by using the computer program TIMING-MSChart [28]. An initial population of 10 newly laid eggs was used for the simulation. The stage growth rate of stage *j* was calculated according to Huang et al. (2018) [29].
$$\phi_{j,t}=\log\left(\frac{n_{j,t+1}+1}{n_{j,t}+1}\right)$$
$$r_{j,t}=\log\left(\frac{n_{j,t+1}+1}{n_{j,t}+1}\right)$$

As the population approaches a stable age-stage distribution, the number of individuals of each stage (*n_j_*_,*t*_) and the total population size (*n*_total,*t*_) will increase at the finite rate of increase (*λ*) and the intrinsic rate of increase (*r*). These can be expressed as:$$\phi_{j,t}=\log\left(\frac{n_{j,t+1}+1}{n_{j,t}+1}\right)\approx \log\left(\frac{\lambda n_{j,t}}{n_{j,t}}\right)=\log\left(\frac{\lambda n_{\mathrm{total},t}}{n_{\mathrm{total},t}}\right)=\log\;\lambda$$
$$r_{j,t}=\log\left(\frac{n_{j,t+1}+1}{n_{j,t}+1}\right)=\ln\left(n_{j,t+1}+1\right)-\ln\left(n_{j,t}+1\right)$$

## 3. Results

### 3.1. Development and Fecundity of P. marginatus after Host Plant Shifting

There were no significant differences in egg duration among the four host plants in the F_0_ generation. However, the developmental times of female and male nymphs fed on *C*. *papaya* were significantly shorter than those on the three other hosts. Extremely long developmental times occurred in both female and male nymphs when fed on *A*. *philoxeroides*. The detailed development durations for each instar are contained in Appendix A. The female adults reared on *C*. *papaya* lived significantly longer than those fed on the other three plants, although there was no significant difference between those fed on *C*. *papaya* and *S*. *tuberosum*. The egg duration was, however, significantly longer in the F_1_ and F_2_ generations when reared on *I*. *batatas* and *S*. *tuberosum*. The durations of male nymphs on *I*. *batatas* and *S*. *tuberosum* were shortened in the F_1_ and F_2_ generations. The durations of the female nymphs were unchanged when reared on the three host plants. The female adult longevities were unchanged on *C*. *papaya* and *I*. *batatas*, but were shortened on *S*. *tuberosum*. The adult longevities of the males were shortened on *I*. *batatas* and *S*. *tuberosum*, but unchanged on *C*. *papaya* (Table 1).

The age-stage life table is capable of describing the stage differentiation; therefore, obvious stage overlapping can be observed. When *P*. *marginatus* individuals were reared on *A*. *philoxeroides* in F_0_, the probability of an egg surviving to the 2nd instar was extremely low (i.e., 0.150, 11 individuals), and significantly lower than on other host plants. Only two eggs successfully developed into female adults (Figure 1). In contrast, the survival rates to the 2nd instar when reared on *C*. *papaya* were as high as 0.924 and 0.930 in F_1_ and F_2_, respectively; higher survival rates to female adulthood (0.489 and 0.490, respectively) were also observed in F_1_ and F_2_. Similar high survival rates occurred in the male adults (0.435 in F_1_ and 0.440 in F_2_). Lower survival rates were observed when reared on *I*. *batatas* and *S*. *tuberosum* (Figure 2). The narrow distribution of male adult survival curves (*s_xj_*) showed that all male adults had shorter lifespans than the females.

The preadult survival rate of *P*. *marginatus* reared on *A*. *philoxeroides* in F_0_ was extremely low (*s_a_* = 0.110), while no significant differences occurred among *C*. *papaya*, *I*. *batatas,* and *S*. *tuberosum*. The preadult survival rate of *P*. *marginatus* reared on *I*. *batatas* increased to 0.933 in F_2_. Higher proportions of female adults of *P*. *marginatus* were observed in F_0_ on *C*. *papaya* (*N_f/_N* = 0.495) and *S*. *tuberosum* (*N_f/_N* = 0.450). The *N_f/_N* value on *I*. *batatas* was 0.170. An extremely low *N_f/_N* value (0.023) was observed on *A*. *philoxeroides*. In the F_2_ generation, the *N_f/_N* values remained constant on *C*. *papaya* and *S*. *tuberosum*, but increased to 0.367 on *I*. *batatas*. In F_0,_ a significantly high proportion of male adults (*N_m/_N*) of *P*. *marginatus* was observed on *I*. *batatas* (0.650). The *N_m/_N* values on *C*. *papaya* and *S*. *tuberosum* were 0.411 and 0.440, respectively. An extremely low *N_m/_N* ratio (0.090) was observed on *A*. *philoxeroides*. The *N_m/_N* ratio did not change from F_1_ to F_2_ (Table 2). 

In the F_0_ generation, the highest fecundity (*F*) of *P*. *marginatus* occurred on *C*. *papaya* (202.70 hatched eggs/female), which was significantly higher than in the other three plants. *Paracoccus marginatus* produced, on average, 6.50 eggs/female when reared on *A*. *philoxeroides*. None of the eggs produced on this host were viable, so the mean fecundity was zero (Table 2). Lower fecundity rates were observed on *I*. *batatas* and *S*. *tuberosum,* with 98.83 hatched eggs/female and 127.46 hatched eggs/female, respectively. On *I*. *batatas*, the fecundity increased in F_1_ (229.50 hatched eggs/female) and F_2_ (203.76 hatched eggs/female) (Table 2).

The age-specific survival rate (*l_x_*) curve is the simplified version of *s_xj_*; thus, the stage differentiation is not observable. The 50% survival rates of *P*. *marginatus* in F_0_ occurred at 26, 25, 26, and 13 d on *C*. *papaya*, *I*. *batatas*, *S*. *tuberosum*, and *A*. *philoxeroides*, respectively (Figure 3). In F_2_, the 50% survival rates of *P*. *marginatus* on *I*. *batatas* and *S*. *tuberosum* changed at 26 and 24 d, respectively (Figure 4). Higher curves of the age-specific fecundity (*m_x_*) and net maternity (*l_x_m_x_*) were observed on *C*. *papaya* in F_0_. Although there was a relatively high peak of 27 eggs at 40 d on *I*. *batatas*, the low survival rate (*l_x_*) caused the net maternity rates (*l_x_m_x_*) to be very low. When reared on *S*. *tuberosum*, the high peak of *m_x_* (18.4 eggs) occurred at 26 d, and the remaining *m_x_* values were, for the most part, greater than 5 eggs (Figure 3). The *m_x_* and *l_x_m_x_* values on *C*. *papaya* and *S*. *tuberosum* did not change significantly; they did, however, increase on *I*. *batatas* during the F_1_ and F_2_ generations (Figure 4).

The life expectancy rates of newly laid eggs of *P*. *marginatus* were 29.0, 24.5, 29.3, and 14.9 d in F_0_. The survival rate from the 1st instar to the 2nd instar on *A*. *philoxeroides* was extremely low (0.15), and individuals surviving to the 2nd instar could, for the most part, complete their development to adults; hence, the *e_xj_* curve of the 2nd instar was significantly higher than in the 1st instar (Figure 5). The detailed *e_xj_* curves on *C*. *papaya*, *I*. *batatas,* and *S*. *tuberosum* during F_1_ and F_2_ are shown in Appendix A.

The age-stage-specific reproductive values (*v_xj_*) at age zero were exactly equal to the finite rates of increase (*λ*), i.e., 1.1945, 1.0970, and 1.1475. The *v_xj_* increased with age. When reared on *I*. *batatas* in F_0_, the *v_xj_* curve significantly increased when female adults emerged. Similar increases in the *v_xj_* curves were observed on *C*. *papaya* and *S*. *tuberosum*; due to the high percentage of female adults, however, this increase was not obvious. The peak dates of *v_xj_* were close to the total preoviposition period (TPOP) (Figure 6). The detailed *v_xj_* curves on *C*. *papaya*, *I*. *batatas,* and *S*. *tuberosum* for F_1_ and F_2_ are shown in Appendix A.

### 3.2. Population Parameters of P. marginatus after Host Plant Shifting

There were significant differences in the population parameters in the F_0_ generation of *P*. *marginatus* after host plant shifting. The highest values of the net reproductive rate (*R*_0_), intrinsic rate of increase (*r*), and finite rate of increase (*λ*) for *P*. *marginatus* occurred on *C*. *papaya* (i.e., 100.26 offspring, 0.1778 d^−1^ and 1.1945 d^−1^). Significantly lower *R*_0_, *r,* and *λ* values were observed when reared on *I*. *batatas* and *S*. *tuberosum*. Only inviable eggs were produced on *A*. *philoxeroides*; thus, the population parameters could not be estimated on this host. The mean generation time (*T*) of *P*. *marginatus* reared on *C*. *papaya* was significantly shorter than on *I*. *batatas* and *S*. *tuberosum*. Although the *R*_0_, *r,* and *λ* values were not significantly changed in the F_1_ and F_2_ individuals when reared on *C*. *papaya* and *S*. *tuberosum*, higher values did occur in the F_1_ and F_2_ generations when reared on *I*. *batatas* (Table 3).

### 3.3. Population Projection of P. marginatus

Starting with an initial population of 10 newly enclosed 1st-instar nymphs, *P*. *marginatus* could develop to the third generation on *C*. *papaya* within 60 d, with a population size reaching as many as 89,552 individuals. However, only two intact generations were observed on *I*. *batatas* and *S*. *tuberosum*, where the population sizes at 60 d were 12,067 and 15,555 individuals, respectively. When the life tables of the 2.5th and 97.5th percentiles of the net reproductive rate (*R*_0_) were used to project the variability of the population growth, the population sizes on *C*. *papaya* ranged from 49,033 to 144,038. However, when the life tables of the 2.5th and 97.5th percentiles of the finite rate of increase (*λ*) were used to project the variability of population growth, the population sizes of *P*. *marginatus* ranged from 47,452 to 131,289 (Figure 7). The growth rate curves of all stages fluctuated around the intrinsic rate of increase (*r*) (Figure 8).

## 4. Discussion

With the intention to improve our understanding in the fitness changes of *P*. *marginatus* after host plant shifting, we investigated the development, fecundity, and population parameters in *P*. *marginatus* within three consecutive generations after being transferred from *S*. *tuberosum* to *C*. *papaya*, *I*. *batatas,* and *A*. *philoxeroides*. In addition, the population growth rates of the insects on *C*. *papaya*, *I*. *batatas,* and *S*. *tuberosum* were projected. The study showed that *P*. *marginatus* transferred to *C*. *papaya* had a higher fitness level. When transferred to *I*. *batatas*, the fitness decreased initially and then recovered after two generations. *Paracoccus marginatus* individuals could rapidly expand their populations on the above host plants. *Alternanthera philoxeroides* was not suitable for the development of *P*. *marginatus*.

Multiple factors such as the population growth and total egg production should be adequately considered when evaluating the fitness of an insect population. The construction and comparison of life tables is the most comprehensive method for describing the population growth, development, survival, and reproduction of a species. Insects of different sexes and stages will usually demonstrate different responses when exposed to variations in their host plants, the numbers and composition of their biological enemies, extreme climate conditions, and pesticides, and consequently it is necessary to take all of these into consideration prior to formulating an effective pest management strategy [30,31,32,33]. In order to accomplish this, life tables are fundamental to achieving a comprehensive assessment of a population’s fitness on a given host plant. Thus, it was important to use the age-stage, two-sex life table method to assessed the fitness changes that occurred after host plant shifting in *P*. *marginatus*.

The age-stage, two-sex life table not only includes the male component of a population, but is also capable of describing the overlapping and differentiation of each stage [29]. Although the males and females of *P*. *marginatus* have different numbers of developmental stages, the stage differentiation can still be precisely described.

The hatch rates of eggs vary with the age of the female adults; hence, using only hatched eggs will enable a more accurate estimate of the population parameters being studied [14,34]. The highest fecundity of *P*. *marginatus* was observed on *C*. *papaya* (*F* = 215.27 hatched eggs/female). Seni et al. (2015) reported the fecundity of *P*. *marginatus* on *C*. *papaya* as 291 total eggs/female (greater than 215.27 hatched eggs/female) [35]; however, the hatch rate was omitted in their study.

By using the age-stage, two-sex life table, He et al. (2021) reported a longer developmental duration and lower intrinsic rate of increase for *Spodoptera frugiperda* (J. E. Smith) (Lepidoptera: Noctuidae) when reared on soybean, while a shorter developmental duration and higher intrinsic rate occurred on sunflower [36]. Karimi-Pormehr et al. (2018) reported a shorter developmental time, higher survival rate, and greater fecundity in *Sitotroga cerealella* (Olivier) (Lepidoptera: Gelechiidae) on a more suitable cultivar (‘19A_1_′) of barley, while noting a longer developmental time and lower fecundity when reared on a less suitable cultivar (‘Fajr30′) [37]. In this study, when *P*. *marginatus* was reared on *C*. *papaya*, the developmental durations of the 1st, 2nd, and 3rd instar (female) individuals were significantly reduced (Appendix A); however, the reverse occurred when reared on *I*. *batatas* and *S*. *tuberosum*. While the fecundity of *P*. *marginatus* was significantly higher on *C*. *papaya* overall in this study, it was significantly lower on *I*. *batatas* (F_0_). The insects have trade-offs between development and reproduction. When the basic ‘development’ need of insects are met by suitable host plants, insects tend to allocate more energy to reproduction.

Our results showed that *P*. *marginatus* reared on *C*. *papaya* had a significantly higher proportion of female adults (*N_f_*/*N*), while a lower *N_f_*/*N* occurred on *I*. *batatas* in the F_0_ generation. Lewontin (1965) demonstrated that the first age of reproduction plays an important role in the values of *r* and *λ* [38]. When *P*. *marginatus* was reared on *C*. *papaya*, reproduction in the F_0_ generation started at 18 d, but advanced to 15 d in F_2_. The three-day change resulted in the value of *r* increasing from 0.1778 d^−1^ (F_0_) to 0.1824 d^−1^ (F_2_), while *λ* increased from 1.1945 d^−1^ (F_0_) to 1.2000 d^−1^ (F_2_) (Figure 4 and Figure 5). Consequently, this change resulted in *P*. *marginatus* reared on *C*. *papaya* having higher values for their population parameters (*r* and *λ*) due to their higher survival and fecundity rates on this host. The opposite was true when reared on *I*. *batatas* and *S*. *tuberosum*.

By using the age-stage, two-sex life table, the stage structure and fluctuations in growth rate in different stages can be observed using population projection. In addition, the life tables constructed based on the 2.5th and 97.5th percentiles of *R*_0_ and *λ* can be used to disclose the variabilities that occur during population growth [29].

When host plant shifting happens, the fitness of the insect population to the new host plant may recover after a few generations. Quezada et al. (2015) showed that *Choristoneura fumiferana* (Clemens) (Lepidoptera: Tortricidae) consecutively reared on less nutritional host plants for three generations would show an adaptive response [39]. Meihls et al. (2008) demonstrated that after three generations of being reared on Bt corn, the survival rate of *Diabrotica v. virgifera* (LeConte) (Coleoptera: Chrysomelidae) was comparable to beetles reared on normal corn [40]. In this study, when *P*. *marginatus* transferred from *S*. *tuberosum* to *C*. *papaya*, all population parameters were significantly higher than on other plants during three generations. This demonstrated that even though *P*. *marginatus* initially survived on *S*. *tuberosum* for multiple generations, the insects transferred to *C*. *papaya* still had a higher fitness level. However, after transferal to *I*. *batatas*, the fitness of *P*. *marginatus* initially decreased in F_0_ and then rebounded in F_1_ and F_2_. *Paracoccus marginatus* showed a higher ability to recover fitness on *I. batata*. Based on the observation that females were unable to produce viable eggs on *A*. *philoxeroides*, we concluded that this host was unsuitable for *P*. *marginatus*. This differences in the fitness of *P. marginatus* to host plants may be due to the volatiles, nutrients of host plants, and so on (unpublished data from the authors).

The age-stage, two-sex life table has been used in a number of studies involving the adaptation of insects on different host plants. Guo et al. (2021) reported that compared with being reared on potato and tobacco, *Spodoptera frugiperda* reared on maize exhibited a shorter developmental time in the larval period, more female individuals, and a higher reproductive rate [41]. Nemati-Kalkhoran et al. (2018) reported the life table characteristics of *Rhyzopertha dominica* (Coleoptera: Bostrichidae) on different barley cultivars, demonstrating that a higher net reproductive rate and intrinsic rate of increase occurred on the cultivar ‘Mahoor’ [42]. Jaleel et al. (2018) reported that *Bactrocera dorsalis* (Diptera: Tephritidae) females produced more eggs on guava than banana [43].

Cipollini and Peterson (2018) pointed out the potential effects of host shifting, including the importance of phytophagous insects being able to find and utilize their ancestral hosts, potentially leading to host range expansions [44]. The present study reports the fitness of *P*. *marginatus* after transferal from *S*. *tuberosum* to *C*. *papaya*, *I*. *batatas,* and *A*. *philoxeroides*. *Ipomoea batatas* and *S*. *tuberosum* are important food crops and *C*. *papaya* is an important fruit [45,46,47]. Our results demonstrate the potential damage of *P. marginatus* to *I*. *batatas* and *S*. *tuberosum*, and again verify the severe damage of *P. marginatus* to *C*. *papaya*, even if the insects transfer from suboptimal host plants. These results indicate that outbreaks of *P. marginatus* are possible in the future, and should they occur may result in serious economic damage. This study provides new insights into the host plant fitness, prevalence, and potential pest control of *P. marginatus*.

## Figures and Tables

**Figure 1 insects-13-00804-f001:**
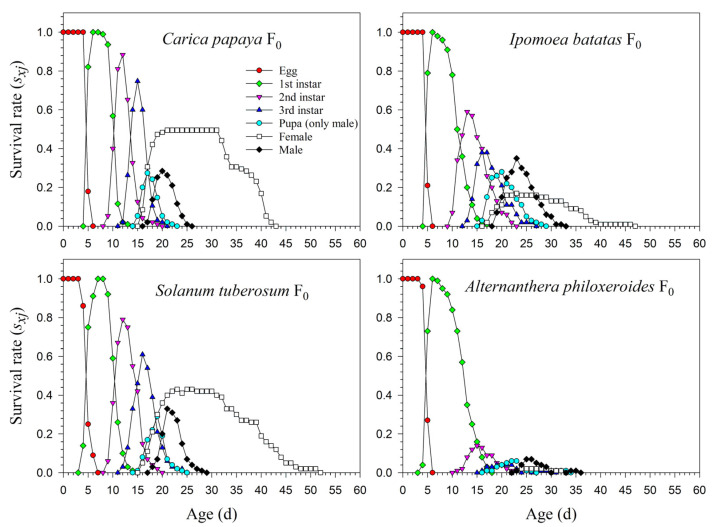
Age-stage-specific survival rates (*s_xj_*) of *Paracoccus marginatus* individuals reared on four different host plants (F_0_).

**Figure 2 insects-13-00804-f002:**
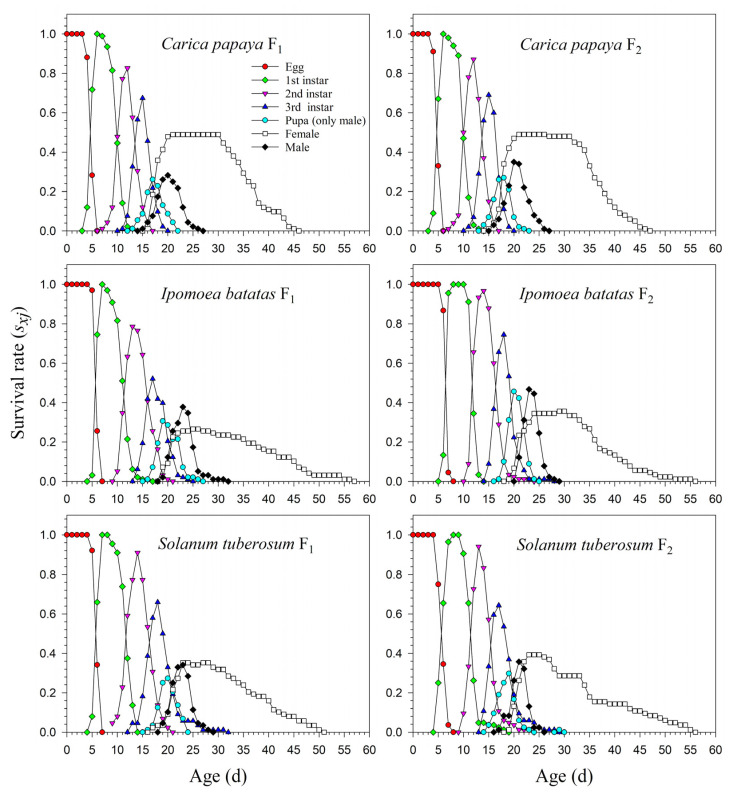
Age-stage-specific survival rates (*s_xj_*) of *Paracoccus marginatus* individuals reared on three different host plants (F_1_–F_2_).

**Figure 3 insects-13-00804-f003:**
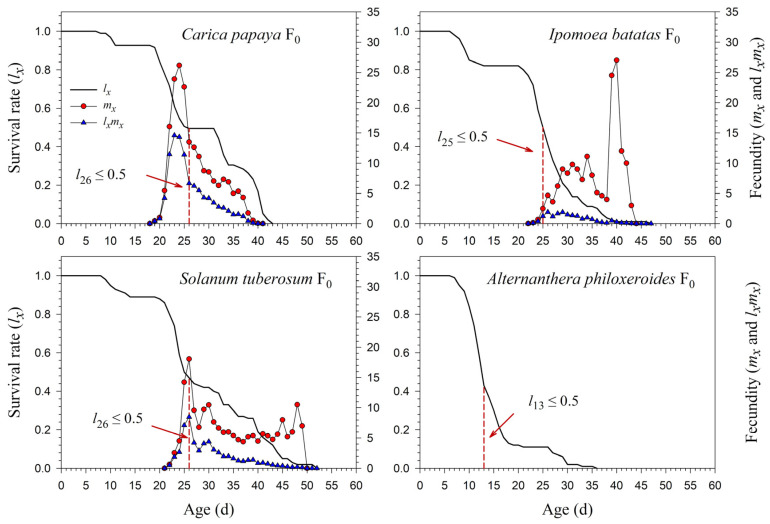
Age-specific survival (*l_x_*), fecundity (*m_x_*), and net maternity (*l_x_m_x_*) rates of *Paracoccus marginatus* individuals reared on four different host plants (F_0_). The red vertical dashed lines in the figure denote the age at which the survival rate *l_x_* ≤ 0.5.

**Figure 4 insects-13-00804-f004:**
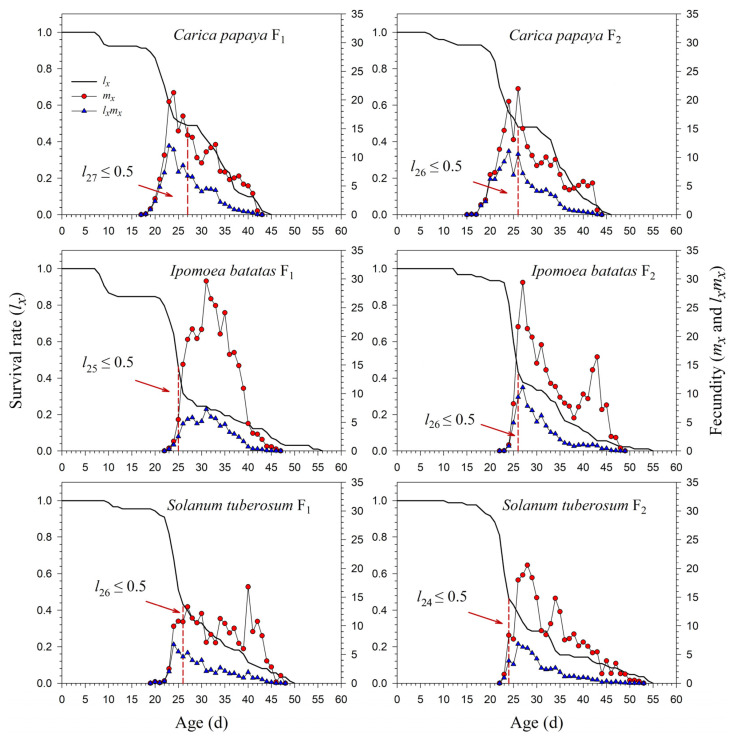
Age-specific survival (*l_x_*), fecundity (*m_x_*), and net maternity (*l_x_m_x_*) rates of *Paracoccus marginatus* individuals reared on three different host plants (F_1_–F_2_). The red vertical dashed lines in the figure denote the age at which the survival rate *l_x_* ≤ 0.5.

**Figure 5 insects-13-00804-f005:**
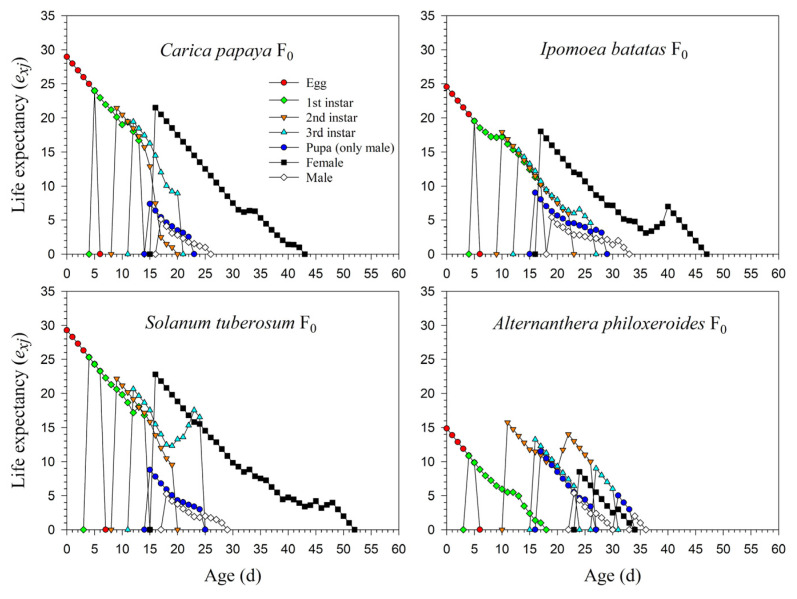
Age-stage-specific life expectancy (*e_xj_*) rates of *Paracoccus marginatus* individuals reared on four different host plants (F_0_).

**Figure 6 insects-13-00804-f006:**
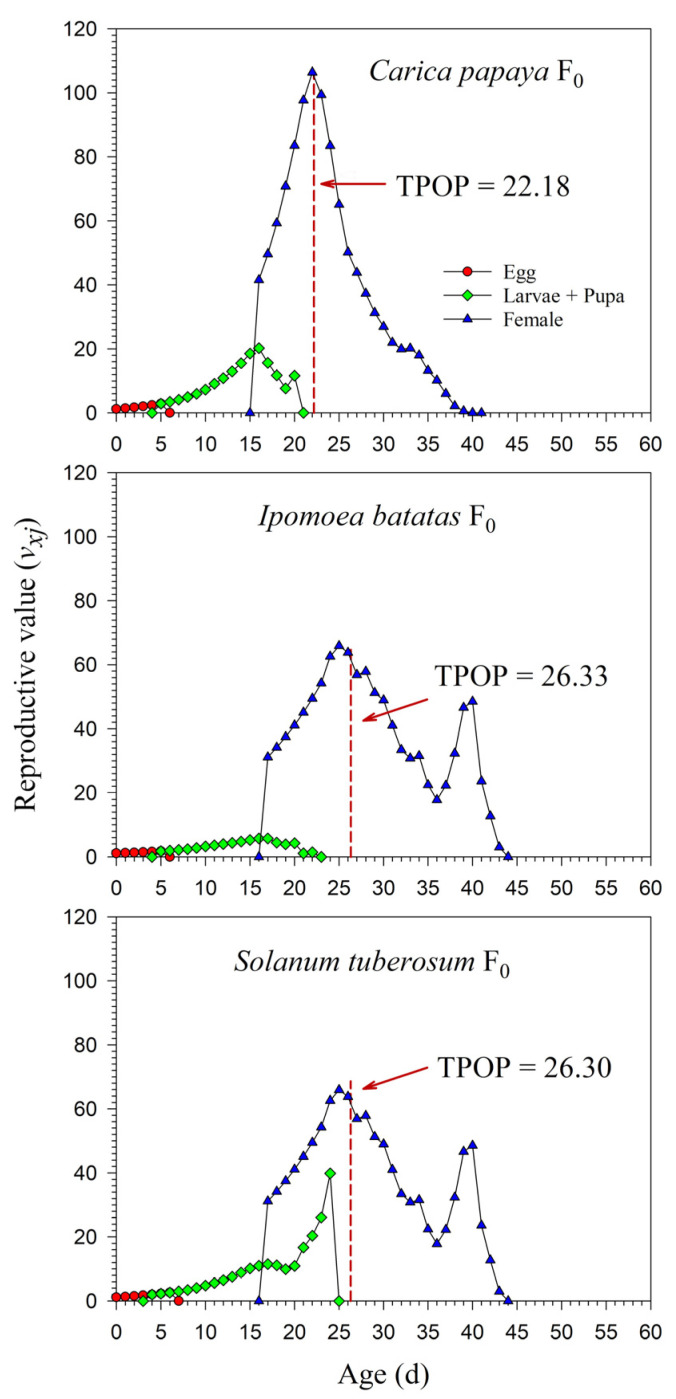
Age-stage-specific reproductive value (*v_xj_*) rates of *Paracoccus marginatus* individuals reared on three different host plants (F_0_). The red vertical dashed line in each figure denotes the total preoviposition period (TPOP).

**Figure 7 insects-13-00804-f007:**
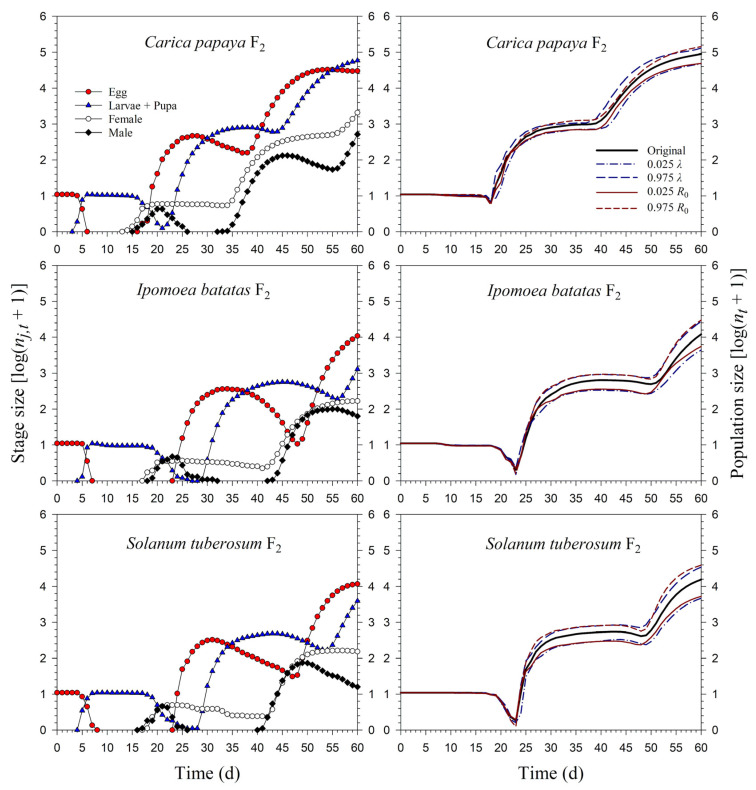
Stage size (log (*n_j_*_,*t*_ + 1)) and population size (log (*n_t_* + 1)) rates of *Paracoccus marginatus* individuals reared on three different host plants (F_2_).

**Figure 8 insects-13-00804-f008:**
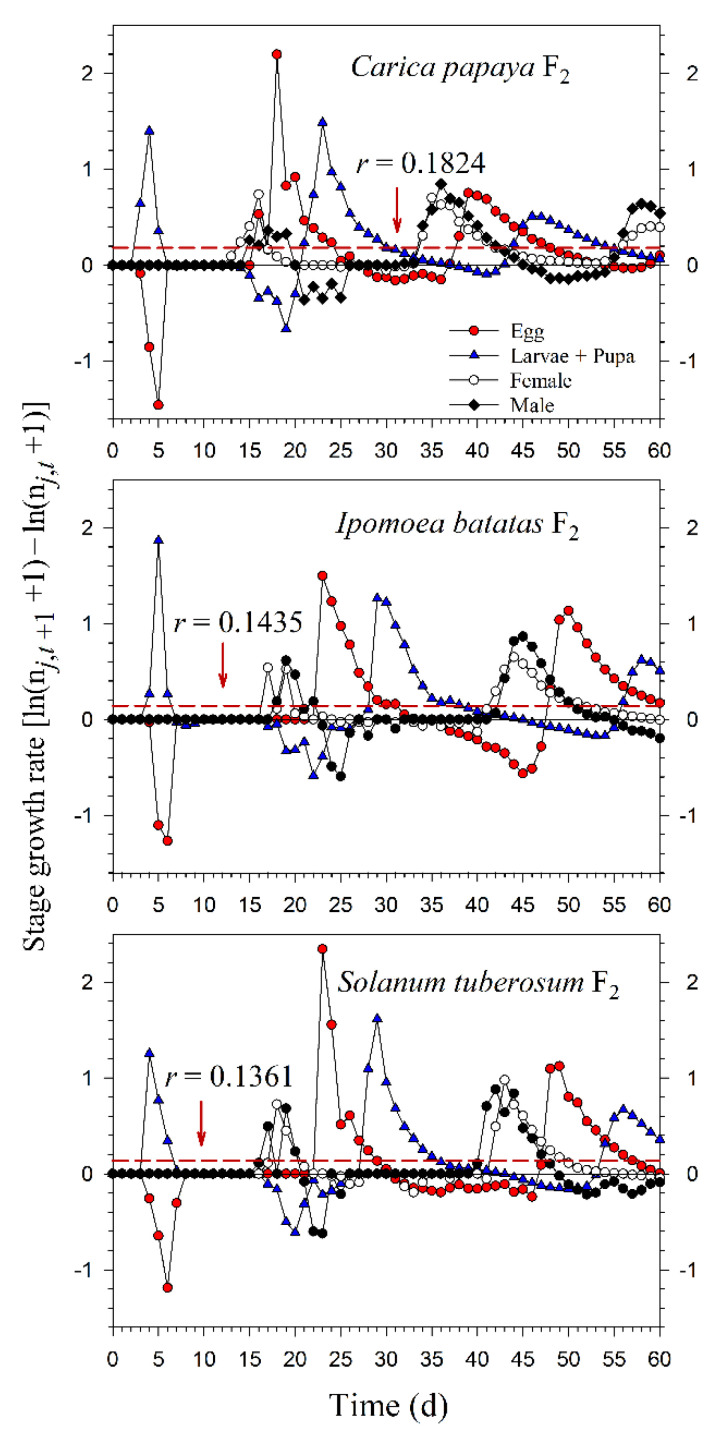
Fluctuations in the growth rates of each life stage of *Paracoccus marginatus* individuals reared on three different host plants (F_2_). The red vertical dashed lines in each figure denote the intrinsic rate of increase (*r*).

**Table 1 insects-13-00804-t001:** The developmental times of *Paracoccus marginatus* individuals reared on four different host plants (F_0_–F_2_).

Stage (d)	Generation	*Carica papaya*		*Ipomoea batatas*		*Solanum tuberosum*		*Alternanthera philoxeroides*	
		*n*	Mean ± SE	*n*	Mean ± SE	*n*	Mean ± SE	*n*	Mean ± SE
Egg	F_0_	95	5.18 ± 0.04 aA	100	5.21 ± 0.04 aC	100	5.20 ± 0.08 aB	100	5.23 ± 0.05 a
	F_1_	92	5.16 ± 0.06 bA	98	6.22 ± 0.05 aB	88	6.26 ± 0.06 aA	-	-
	F_2_	100	5.24 ± 0.06 cA	90	6.91 ± 0.04 aA	84	6.13 ± 0.09 bA	-	-
Female nymph	F_0_	47	12.36 ± 0.16 cA	17	14.29 ± 0.41 bA	45	14.04 ± 0.30 bA	2	18.50 ± 0.50 a
	F_1_	45	11.98 ± 0.18 bA	30	14.00 ± 0.31 aA	40	15.07 ± 0.47 aA	-	-
	F_2_	49	12.24 ± 0.16 bA	33	13.70 ± 0.34 aA	40	14.32 ± 0.35 aA	-	-
Male nymph	F_0_	39	13.92 ± 0.21 dA	65	17.18 ± 0.30 bA	44	15.43 ± 0.20 cA	9	20.22 ± 1.05 a
	F_1_	40	13.50 ± 0.26 bA	53	15.53 ± 0.24 aB	44	15.20 ± 0.19 aAB	-	-
	F_2_	44	13.64 ± 0.21 cA	51	15.55 ± 0.15 aB	38	14.61 ± 0.24 bB	-	-
Female adult	F_0_	47	20.09 ± 0.55 aA	17	15.65 ± 1.23 bA	45	19.87 ± 0.94 abA	2	8.00 ± 1.00 c
	F_1_	45	19.18 ± 0.56 aA	30	18.53 ± 1.63 aA	40	13.97 ± 1.27 bB	-	-
	F_2_	49	19.67 ± 0.56 aA	33	17.12 ± 1.22 aA	40	14.28 ± 1.44 bB	-	-
Male adult	F_0_	39	3.28 ± 0.18 aA	65	3.48 ± 0.14 aA	44	3.41 ± 0.20 aA	9	3.67 ± 0.62 a
	F_1_	40	3.48 ± 0.17 aA	53	3.21 ± 0.14 aAB	44	3.11 ± 0.15 aAB	-	-
	F_2_	44	3.50 ± 0.16 aA	51	3.00 ± 0.13 bB	38	2.87 ± 0.13 bB	-	-

The data (means ± SE) followed by the same letters were not significantly different as assessed by paired bootstrap test (*p* < 0.05). The lowercase letters in the same row indicated comparisons among different host plants in the same generations, and the capital letters in the same column indicated comparisons among different generations on the same host plants. The 1st instar, 2nd instar, 3rd instar, and pupae samples were combined into nymph samples for the data statistics.

**Table 2 insects-13-00804-t002:** Preadult survival rates (*s_a_*), proportions of female adults (*N_f_/N*), proportions of male adults (*N_m_*/*N*), and fecundity rates (*F*) of *Paracoccus marginatus* individuals reared on four different host plants (F_0_–F_2_).

Population Parameter	Generation	*Carica papaya*	*Ipomoea batatas*	*Solanum tuberosum*	*Alternanthera philoxeroides*
Preadult survival rate (*s_a_*)	F_0_	0.905 ± 0.030 aA	0.820 ± 0.038 aB	0.890 ± 0.031 aA	0.110 ± 0.031 b
	F_1_	0.924 ± 0.028 abA	0.847 ± 0.036 bB	0.955 ± 0.022 aA	-
	F_2_	0.930 ± 0.026 aA	0.933 ± 0.026 aA	0.929 ± 0.028 aA	-
Proportion of female adults (*N_f/_N*)	F_0_	0.495 ± 0.051 aA	0.170 ± 0.038 bB	0.450 ± 0.050 aA	0.023 ± 0.012 c
	F_1_	0.489 ± 0.052 aA	0.306 ± 0.047 bA	0.455 ± 0.053 aA	-
	F_2_	0.490 ± 0.050 aA	0.367 ± 0.051 aA	0.476 ± 0.055 aA	-
Proportion of male adults (*N_m_*/*N*)	F_0_	0.411 ± 0.050 bA	0.650 ± 0.048 aA	0.440 ± 0.049 bA	0.090 ± 0.050 c
	F_1_	0.435 ± 0.052 aA	0.541± 0.050 aA	0.500 ± 0.053 aA	-
	F_2_	0.440 ± 0.049 aA	0.567 ± 0.052 aA	0.452 ± 0.054 aA	-
Fecundity (*F*) (hatch eggs/female)	F_0_	202.70 ± 17.68 aA	98.93 ± 23.04 bB	127.46 ± 18.79 bA	0
	F_1_	202.46 ± 24.08 aA	229.50 ± 41.84 aA	127.38 ± 25.40 bA	-
	F_2_	215.27 ± 20.32 aA	203.76 ± 35.72 abA	121.51 ± 25.35 bA	-

The data (means ± SE) followed by the same letters were not significantly different as assessed by paired bootstrap test (*p* < 0.05). The lowercase letters in the same row indicated comparisons among different host plants in the same generations, and the capital letters in the same column indicated comparisons among different generations on the same host plants.

**Table 3 insects-13-00804-t003:** Population parameters of *Paracoccus marginatus* individuals reared on three different host plants (F_0_–F_2_).

Population Parameter	Generation	*Carica papaya*	*Ipomoea batatas*	*Solanum tuberosum*
Net reproductive rate (*R*_0_) (offspring)	F_0_	100.26 ± 13.56 aA	16.82 ± 5.31 cB	57.36 ± 10.53 bA
	F_1_	98.97 ±15.76 aA	70.32 ±16.63 abA	57.98 ± 13.31 bA
	F_2_	105.51 ± 4.98 aA	74.64 ±5.09 abA	57.88 ±13.74 bA
Intrinsic rate of increase (*r*) (d^−1^)	F_0_	0.1778 ± 0.0053 aA	0.0926 ± 0.0109 cB	0.1376 ± 0.0069 bA
	F_1_	0.1747 ± 0.0064 aA	0.1357 ± 0.0080 bA	0.1389 ± 0.0084 bA
	F_2_	0.1824 ± 0.0064 aA	0.1435 ± 0.0075 bA	0.1361 ± 0.0085 bA
Finite rate of increase (*λ*) (d^−1^)	F_0_	1.1945 ± 0.0064 aA	1.0970 ± 0.0119 cB	1.1475 ± 0.0079 bA
	F_1_	1.1909 ± 0.0076 aA	1.1454 ± 0.0091 bA	1.1490 ± 0.0097 bA
	F_2_	1.2000 ± 0.0076 aA	1.1543 ± 0.0086 bA	1.1459 ± 0.0097 bA
Mean generation time (*T*) (d)	F_0_	25.92 ± 0.30 bA	30.48 ± 1.08 aAB	29.43 ± 0.61 aA
	F_1_	26.30 ± 0.46 cA	31.34 ± 0.42 aA	29.23 ± 0.58 bA
	F_2_	25.56 ± 0.40 bA	30.05 ± 0.39 aB	29.81 ± 0.61 aA

The data (means ± SE) followed by the same letters were not significantly different as assessed by paired bootstrap test (*p* < 0.05). The lowercase letters in the same row indicated comparisons among different host plants in the same generations, and the capital letters in the same column indicated comparisons among different generations on the same hosts.

## Data Availability

Data are contained within the article or Appendix A.

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
