# Peer review of "Fitness of the Papaya Mealybug, Paracoccus marginatus (Hemiptera: Pseudococcidae), after Transferring from Solanum tuberosum to Carica papaya, Ipomoea batatas, and Alternanthera philoxeroides"

_insects, 2022, doi:10.3390/insects13090804_

Round 1

Reviewer 1 Report

Minor corrections as sticky notes on the pdf.

Author Response

Response to Reviewer 1 Comments

Thank you for the understanding of our work and the valuable comments. We have rewritten some parts of the text to make it clearer and more complete, and the missing information has also been added in the revised manuscript.

A point-by-point response are listed below and shown in red (some similar questions are revised together). We hope these revisions successfully address your concerns and requirements and hope that this manuscript will be accepted. Looking forward to hearing from you soon.

Point 1: 33: please change the words so on to other plant parts

Response 1: We really appreciate your efforts and comments on our manuscript. We have changed the words ‘so on’ to ‘other plant parts’.

Point 2: 42: please delete the words ‘a worst weed,’

Response 2: Thanks a lot for your kind suggestion. We have deleted the words ‘a worst weed,’.

Point 3: 52: please change the words versus old host plant to versus their old host plant

Response 3: Thanks for your valuable suggestion. We have changed the words ‘versus old host plant’ to ‘versus their old host plant’.

Point 4 (The similar questions are revised together):

160: please change the word larvae to nymphs

162: please change the word larvae to nymphs

295: please change the word larvae to nymphs

Response 4: Thanks for your kind comments. We have changed the word ‘larvae’ in these three sentences to ‘nymphs’. Further, we have checked the full text and modified other four sentences with similar problems (the modified text: lines 167, 173, 176, and 183).

Point 5: 170: please change the words larvae durations of females to duration of the female nymphs

Response 5: Thank you very much for your comments. We have changed the words ‘larvae durations of females’ to ‘duration of the female nymphs’.

Point 6: 387: please change the word survival to survived

Response 6: Thanks a lot for your comments. We have changed the word ‘survival’ to ‘survived’.

Point 7: 409: please change the word reported to reports

Response 7: Thanks a lot for your valuable comments. We have changed the word ‘reported’ to ‘reports’.

Other Change

Response 1: We have added a sentence at the end of the Discussion to further explain the significance of this study (the modified text: lines 419-420). The details are as follows:

This study provides new insights into host plant fitness, prevalence, and potential pest control of P. marginatus.

Reviewer 2 Report

An interesting paper with a nice approach to life tables. The work is concisely written and nicely framed.

Some minor points to the language could improve the work as can be seen below, I also feel that a succinct testable hypothesis in the introduction and reference back to this in the discussion would improve this work.

Please see comments as they appear below:

72-78: the introduction aims could be improved by setting the scene with a testable hypothesis.

157: Delete the first sentence, refer to (Figure 1) in parenthesis after statement of results, rather than introducing the tables at the start.

179:  As above, refer to tables in parenthesis after the statement of results. Rather than introduce the tables in the text.

180: Don’t start a sentence with Because.

305 – 307: Delete first sentence put (Figure 8) at the end of the last sentence

316: Add a summary statement, regarding how your hypothesis was supported - see first comment. 

321: don’t start a sentence with because

335: no need to refer to figures from within the discussion

337: Don’t start a paragraph with because

381: “Adaptive response” quotation marks not needed

Author Response

Response to Reviewer 2 Comments

Thank you for the understanding of our work and the valuable comments. We have rewritten some parts of the text to make it clearer and more complete, and the missing information has also been added in the revised manuscript.

A point-by-point response are listed below and shown in red (some similar questions are revised together). We hope these revisions successfully address your concerns and requirements and hope that this manuscript will be accepted. Looking forward to hearing from you soon.

Point 1: 72-78: the introduction aims could be improved by setting the scene with a testable hypothesis.

Response 1: We really appreciate your efforts and comments on our manuscript. We have added testable hypotheses at the end of the Introduction (the modified text: lines 71-78). The details are as follows:

We hypothesized that the fitness of P. marginatus would change after host plant shifting. We also predicted that P. marginatus can readily adapt to some new host plants within a few generations. To test these hypotheses, we measured the development, fecundity, and population parameters in P. marginatus over a span of three consecutive generations after transferred from S. tuberosum to C. papaya, I. batatas and A. philoxeroides. Further, we projected the population growth of insects on C. papaya, I. batatas and S. tuberosum in the F2 generation.

Point 2 (The similar questions are revised together):

157: Delete the first sentence, refer to (Figure 1) in parenthesis after statement of results, rather than introducing the tables at the start.

179: As above, refer to tables in parenthesis after the statement of results. Rather than introduce the tables in the text.

305 – 307: Delete first sentence put (Figure 8) at the end of the last sentence

Response 2: Thanks a lot for your kind comments. We have adjusted the refer to figures/tables of these three sentences to the end of the results. Further, we have checked the full text and modified other five paragraphs with similar problems (the modified text: lines 226-238, 247-253, 257-264, 271-282, and 288-299).

Point 3 (The similar questions are revised together):

180: Don’t start a sentence with Because.

321: don’t start a sentence with because

337: Don’t start a paragraph with because

Response 3: Thank you very much for your comments. We have modified these three sentences starting with Because. Further, we have checked the full text and modified other six sentences with similar problems (the modified text: lines 108-110, 214-215, 226-227, 248-251, 276-277, and 412-413).

Point 4:

316: Add a summary statement, regarding how your hypothesis was supported - see first comment.

Response 4: Thanks for your efforts and valuable comments. We have added a summary statement at the beginning of the Discussion (the modified text: lines 309-319). The details are as follows:

With the intention to improve our understanding in the fitness change of P. marginatus after host plant shifting, we investigated the development, fecundity, and population parameters in P. marginatus within three consecutive generations after transferred from S. tuberosum to C. papaya, I. batatas and A. philoxeroides. In addition, the population growth of insects on C. papaya, I. batatas and S. tuberosum were projected. The study showed that P. marginatus transferred to C. papaya had a higher fitness. When transferred to I. batatas, the fitness decreased initially and then recovered after two generations. Paracoccus marginatus could rapidly expand their populations on the above host plants. Alternanthera philoxeroides was not suit for the development of P. marginatus.

Point 5: 335: no need to refer to figures from within the discussion

Response 5: Thanks a lot for your kind suggestion. We have deleted the refer to figures in the discussion.

Point 6: 381: “Adaptive response” quotation marks not needed

Response 6: Thanks for your kind comments. We have deleted the quotation marks.

Other Change

Response 1: We have added a sentence at the end of the Discussion to further explain the significance of this study (the modified text: lines 419-420). The details are as follows:

This study provides new insights into host plant fitness, prevalence, and potential pest control of P. marginatus.
